# Deep transfer learning artificial intelligence accurately stages COVID-19 lung disease severity on portable chest radiographs

**Jocelyn Zhu, Beiyi Shen, Almas Abbasi, Mahsa Hoshmand-Kochi, Haifang Li, Tim Q. Duong**⊙*

Department of Radiology, Renaissance School of Medicine, Stony Brook University, Stony Brook, New York, United States of America

* tim.duong@stonybrook.edu

**Data Availability Statement:** The complete dataset can be found at www.kaggle.com/dataset/89ab2075b0a23026a0a1dda064441a6d05d38dc1e71b5d04d2c3e5a0196b89a8.

## Abstract

This study employed deep-learning convolutional neural networks to stage lung disease severity of Coronavirus Disease 2019 (COVID-19) infection on portable chest x-ray (CXR) with radiologist score of disease severity as ground truth. This study consisted of 131 portable CXR from 84 COVID-19 patients (51M 55.1±14.9yo; 29F 60.1±14.3yo; 4 missing information). Three expert chest radiologists scored the left and right lung separately based on the degree of opacity (0–3) and geographic extent (0–4). Deep-learning convolutional neural network (CNN) was used to predict lung disease severity scores. Data were split into 80% training and 20% testing datasets. Correlation analysis between AI-predicted versus radiologist scores were analyzed. Comparison was made with traditional and transfer learning. The average opacity score was 2.52 (range: 0–6) with a standard deviation of 0.25 (9.9%) across three readers. The average geographic extent score was 3.42 (range: 0–8) with a standard deviation of 0.57 (16.7%) across three readers. The inter-rater agreement yielded a Fleiss' Kappa of 0.45 for opacity score and 0.71 for extent score. AI-predicted scores strongly correlated with radiologist scores, with the top model yielding a correlation coefficient ($R^2$) of 0.90 (range: 0.73–0.90 for traditional learning and 0.83–0.90 for transfer learning) and a mean absolute error of 8.5% (ranges: 17.2–21.0% and 8.5%-15.5, respectively). Transfer learning generally performed better. In conclusion, deep-learning CNN accurately stages disease severity on portable chest x-ray of COVID-19 lung infection. This approach may prove useful to stage lung disease severity, prognosticate, and predict treatment response and survival, thereby informing risk management and resource allocation.

## Introduction

Coronavirus Disease 2019 (COVID-19) is an infectious disease that can cause severe respiratory illness [1, 2]. First reported in Wuhan, China in December 2019 [3], COVID-19 was declared a pandemic on Mar 11, 2020 (https://www.who.int). More than 2 million people have been infected and more than 120,000 have died of COVID-19 (https://coronavirus.jhu.edu,

**Funding:** unfunded studies.

**Competing interests:** The authors have declared that no competing interests exist.

Apr 14, 2020). These numbers are predicted to continue to rise in the foreseeable future with a high likelihood of a second wave and recurrence. COVID-19 has already overwhelmed many hospitals in many countries.

Portable chest x-rays (CXR) has become an indispensable imaging tool to imaging patients with contagious diseases because it is informative and the equipment can be readily disinfected. CXR provides both the extent and severity of lung infection, and is widely used to quantify disease severity in many lung diseases [4]. The hallmarks of COVID-19 lung infection on CXR include bilateral, peripheral hazy opacity and airspace consolidation [5]. Given the anticipated shortage of intensive care unit (ICU) beds and mechanical ventilators in many hospitals, CXR has the potential to play a critical role in decision-making to determine which patients to put on a mechanical ventilator, monitor disease progression and treatment effects during mechanical ventilation, and determine when it is safe to extubate. Similarly, computed tomography (CT), which offer better sensitivity than CXR, has been used in imaging suspected or confirmed COVID-19 patients, albeit mostly in China early on in the pandemic [6–9]. However, CT is less convenient, not practical in ICU setting, and prone to cross contamination and, thus, it is not widely used in COVID-19 infection or similar contagious disease in the United States.

Deep-learning artificial intelligence (AI) has become increasingly popular for analyzing diagnostic images [10, 11]. AI has the potential to facilitate disease diagnosis, diseases severity staging, and longitudinal monitoring of disease progression. One common machine-learning algorithm is the convolutional neural network (CNN) [12, 13], which takes an input image, learns important features in the image such as size or intensity, and saves these parameters as weights and bias to differentiate types of images [14, 15]. CNN architecture is ideally suited for analyzing images.

The majority of machine learning algorithms to date are trained to solve specific tasks, working in isolation. Models have to be rebuilt from scratch if the feature-space distribution changes. Transfer learning overcomes the isolated learning paradigm by utilizing knowledge acquired for one task to solve related ones. Transfer learning in AI is particularly important for small sample size data because the pre-trained weights enables more efficient training and better performance [16, 17].

There are many AI algorithms that have already been developed for CXR applications (see review [18]) and these AI algorithms can be repurposed to study COVID-19 lung infection. A few studies have already reported AI applications to classify COVID-19 versus non-COVID-19 lung infection using CXR [19–22] and CT [23–26]. Many of these referenced publications here are non-peer reviewed pre-prints. These studies aimed to classify COVID-19 versus non-COVID-19 images, not to stage lung disease severity. Classification of COVID-19 versus non-COVID-19 images for the purpose of diagnosis made by AI unfortunately has poor specificity. The reverse-transcription polymerase chain reaction test of a nasopharyngeal or oropharyngeal swab is still needed for definitive diagnosis. By contrast, deep-learning AI is well suited to stage disease severity. To our knowledge, there have been no studies to date that use deep-learning AI to stage disease severity on CXR of COVID-19 lung infection.

The goal of this study was to employ deep-learning CNN to stage disease severity of COVID-19 infection on CXR (as opposed to classify COVID-19 versus non-COVID-19 lung infection). The ground truths were the disease severity scores determined by expert chest radiologists. Comparison was made with traditional and transfer learning. This approach has the potential to provide frontline physicians an efficient and accurate means to triage patients, assess risk, allocate resources, as well as to monitor disease progression and treatment response. This approach is timely and urgent because of the anticipated widespread shortage of ICU beds and mechanical ventilators.

## Materials and methods

### Data sources

The diagram of patient selection and experimental design is shown in Fig 1. This is a retrospective study using publicly available de-identified data. Images were downloaded from https://github.com/ieee8023/covid-chestxray-dataset [27] on Mar 30, 2020. The original download contained 250 images of COVID-19 and SARS. Only anterior-posterior and posterior-anterior

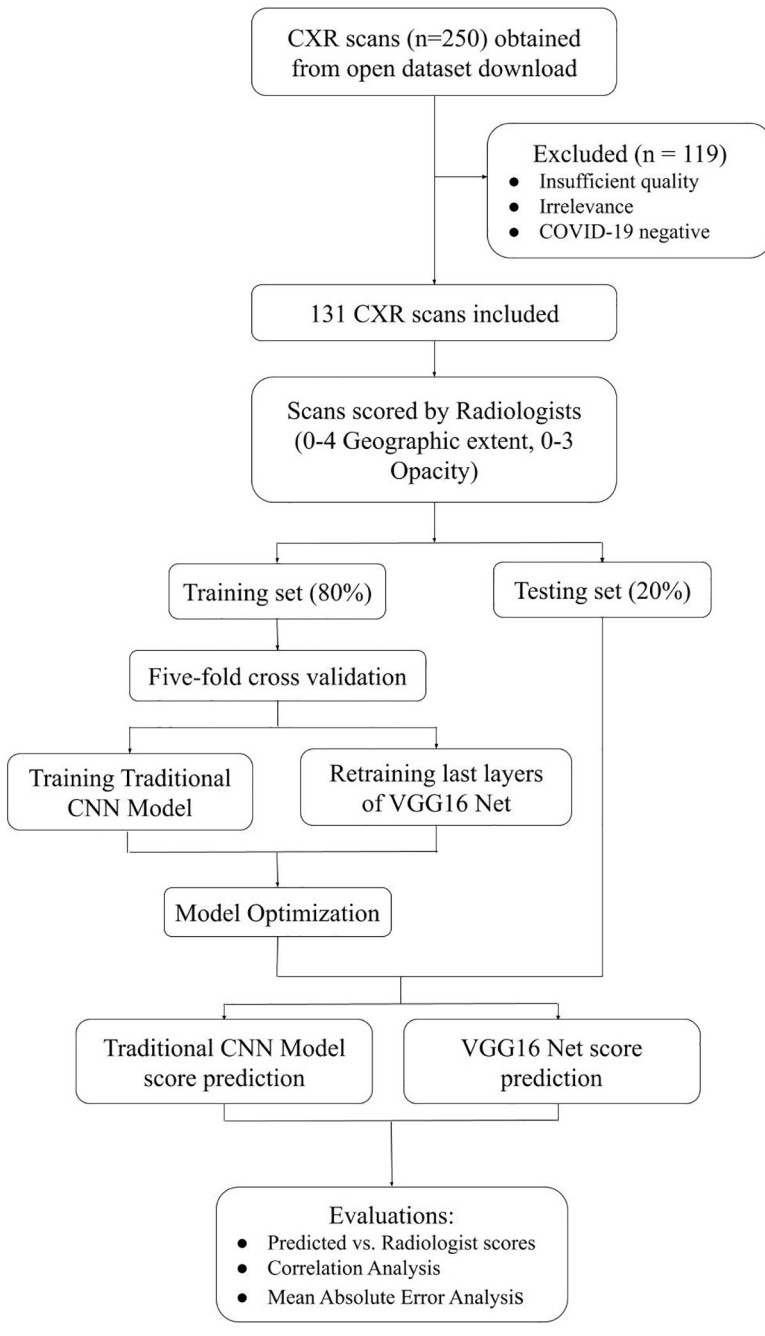

**Fig 1. Diagram of patient selection and experimental design.**

CXR from COVID-19 were included in this study, resulting in final sample size of 131 CXRs from 84 COVID-19 positive patients (51 males and 29 females; 4 with missing information). The mean and standard deviation of age was 55.1±14.9 years old for men and 60.1±14.3 years old for women.

## Radiologist scoring

Radiological scoring is widely used to stage disease severity in many lung diseases [4]. To establish the disease severity score, a group of 6 chest radiologists with at least 20 years of experience worked together to reach consensus by evaluating two dozen images of portable CXRs of COVID-19 patients from Stony Brook University Hospital (these testing CXRs were not used in this study). Our scoring system was adapted from those by Warren et al. [28] and Wong et al [5]. Two board-certified chest radiologists with 20+ years of experience and one radiology resident reviewed image quality and scored the CXR for disease severity using the following criteria based on degree of opacity and geographical extent. The degree of opacity score of 0–3 was assigned to each of the right and left lung as: 0 = no opacity; 1 = ground glass opacity; 2 = consolidation; 3 = white-out. The right and left lung were scored separately and were added together. The geographical extent score of 0–4 was assigned to each of the right and left lung depending on the extent of involvement with ground glass opacity or consolidation: 0 = no involvement; 1 = <25%; 2 = 25–50%; 3 = 50–75%; 4 = >75% involvement. The right and left lung were scored separately and were added together. Sum of the two types of scores (0–14) and the product of the two types scores (0–48) were computed. The average of all radiologist scores were used for analysis.

## CNN

A convolutional neural network (Fig 2A) [12, 13] with an additional regression layer was utilized to predict disease severity scores of CXR on a graded scale. The images were normalized, converted to RGB images, resized to 64x64 pixel images, and separated into 80% training and 20% testing datasets. This model architecture included Convolutional, Activation, Batch Normalization, Max Pooling, Dense layers, and a final Dense regression layer to fit the model to predict continuous values. Specifically, for convolutional neural networks, deep learning models are well-suited to learn complex functions, and optimize performance of regularization methods. Batch normalization layers served to stabilize the learning process by standardizing inputs, and standard rectified linear activation layers for CNNs were used to optimize performance. In general, the number of layers and nodes depends heavily on the input data and prediction goals, therefore heuristics were used to generate a standard CNN model for image analysis, the model was then fine-tuned based on performance on the testing dataset. This resulted in three 3x3 kernel sized convolutional layers. L2 regularization and Dropout layers, using a parametric probability of 0.3 as determined by analyzing performance results, were used to prevent overfitting. An optimal batch size of 8 was determined by five-fold cross validation, and the model was trained for 110 epochs. The loss function was measured by mean squared error as the model predicted continuous values, and the learning rate was set to 0.001, as it proved to perform more positively than the standard 0.01 rate in early training, and the Adam optimizer proved the most successful in minimizing loss.

## Transfer learning

Given the limited dataset, a transfer learning method was also explored to optimize prediction. A VGG16 model was loaded with sample weights trained off the ImageNet dataset [29]. To be compatible with VGG16, the data was normalized, converted to RGB images, and resized into

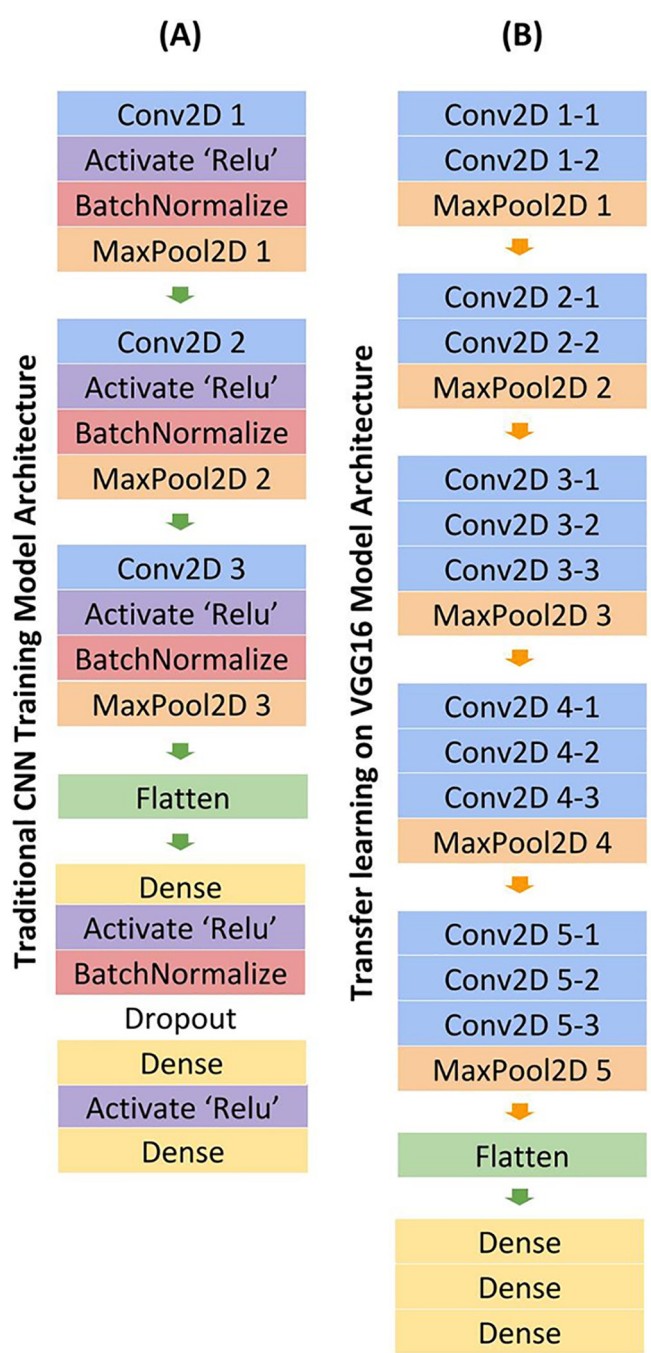

**Fig 2.** CNN Model architecture for traditional training (A) and transfer learning method (B). Model takes in a CXR image as an input. The outputs of the prediction were different severity scores.

224x224 pixel images. The dataset was also split into 80% training and 20% testing. Only the top layers were trained to prevent overfitting, and all convolutional layers were frozen. Standard VGG16 architecture was utilized, with the exception of the fully connected layers (Fig 2B). Once out of the convoluted layers, the model output was flattened, then passed through three Dense layers, including a regression Dense layer. For this transfer learning method, cross validation revealed an optimal batch size of 8 and only 10 epochs were necessary to train the

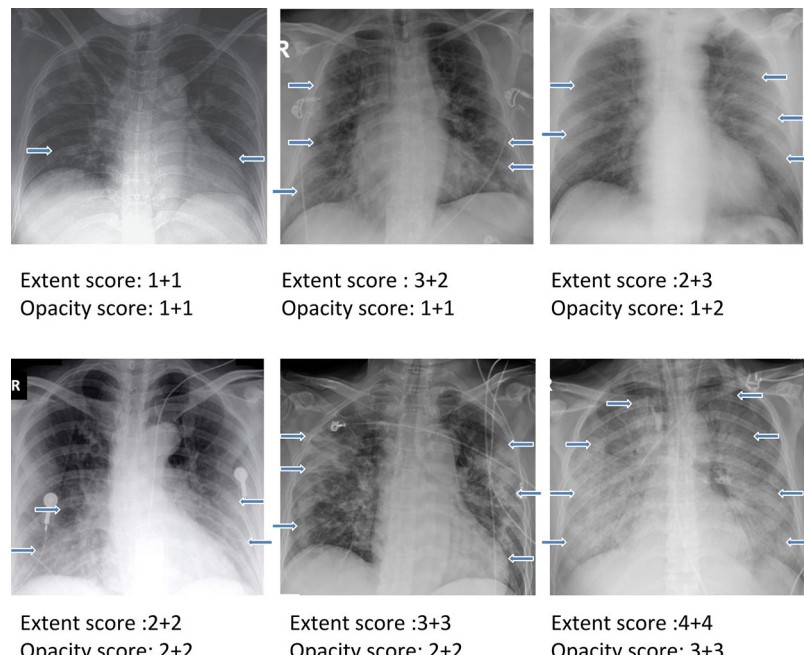

**Fig 3. Examples showing a range of severity scores which include extent and opacity scores.** The two numbers for each are for the right and left lung. Arrows indicate region of Ground glass opacities or/and airspace consolidations.

data, as opposed to the 110 needed for the traditional training method. Adam optimizer and 0.001 learning rate were similarly employed in this model.

### Statistical methods and performance evaluation

The means and standard deviations of the radiologist scores, and the Fleiss' Kappa inter-rater agreement were calculated. Correlation analysis between AI-predicted versus radiologist severity scores were analyzed with slopes, intercepts, correlation coefficients ($R^2$), p-values, and mean absolute errors (MAE) reported. Results using traditional and transfer learning were compared. All results are reported as means ± standard deviations. A $p < 0.05$ was taken to be statistically significant unless otherwise specified. Note that the prediction of graded values precluded receiver operating curve (ROC) analysis.

### Results

The final sample size consisted of 131 CXR from 84 COVID-19 positive patients (51M and 29F; 4 missing information). The men were on average 55.1±14.9 years old and women were 60.1±14.3 years old. CXR examples demonstrating a range of severity scores which include extent and opacity score are illustrated in Fig 3. CXR of COVID-19 positive patients showed hazy opacities and/or airspace consolidation. It has a predominance of bilateral, peripheral

**Table 1. Individual, sum and product of degree of opacity scores and geographic extent scores.** Values in parentheses are ranges for the corresponding scores.

|  | Geographic extent score (0–8) | Opacity score (0–6) | Sum (0–14) | Product (0–48) |
|---|---|---|---|---|
| **Mean** | 3.41 | 2.52 | 5.84 | 11.86 |
| **Standard deviation across 3 readers** | 0.78 | 0.25 | na | na |
| Fleiss' κ | 0.45 | 0.71 | na | na |

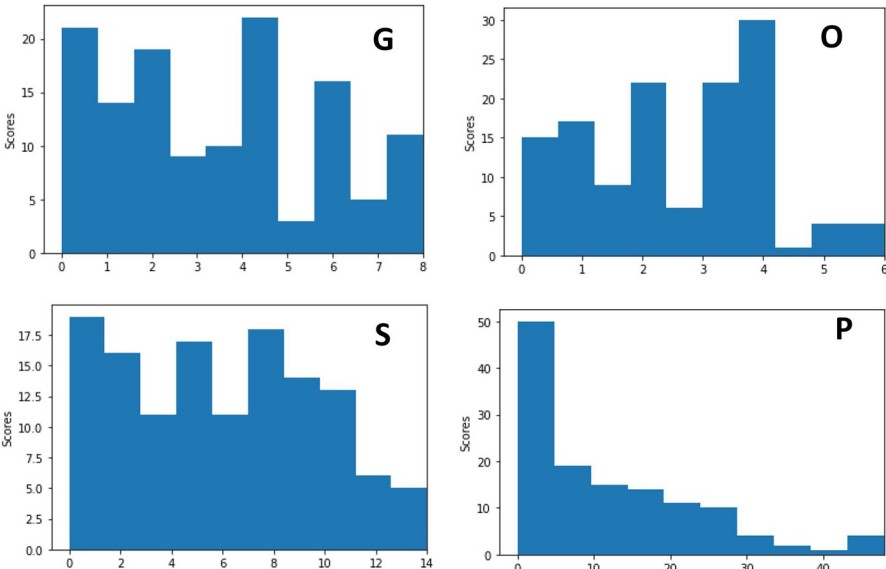

**Fig 4. Histograms of radiologist severity scores.** G: geographic extent, O: degree of opacity, S: sum of the two score types, P: product of the two score types.

and lower lung zone distribution. The results of three raters are summarized in Table 1. The mean opacity score was 2.52 (0–6), with a standard deviation of 0.25 (9.9%) across three readers. The mean geographic extent score was 3.42 (a range from 0 to 8), with a standard deviation of 0.57 (16.7%) across three readers. The low standard deviations across three raters suggest good agreement. Fleiss' Kappa for inter-rater agreement was 0.45 (z = 24.4) for opacity score and 0.71 (z = 29.5) for geographic extent score, indicating good to excellent agreement. The mean of sum of scores was 5.84 (0–14) and mean of product of scores was 11.86 (0–48). Fig 4 shows the histograms of radiologist scores. The product score was slightly skewed toward lower values because numbers multiplied by a number close to zero tended to be of lower values.

The data were separated into 80% training and 20% testing datasets. The loss function decreased and prediction accuracy improved with increasing epochs for both training and validation datasets. For the traditional training method, the loss function typically converged at training epochs of 30–50 epochs, and five-fold validation confirmed optimal hyperparameters of training on 110 epochs. For the transfer learning method employing a VGG16 model, 10 epochs were sufficient to train the model with high predictive ability; more epochs led to overfitting and increased computational time.

Fig 5 shows the AI-predicted scores versus radiologist severity scores of the "test" dataset, obtained for different types of scores using traditional and transfer learning. Overall, there were strong correlations between the AI-predicted and radiologist scores. The slopes, intercepts, $R^2$, p values, and mean absolute errors (MAE) with traditional and transfer learning are summarized in Table 2. The slopes were generally (but not consistently across different scores) closer to unity, the intercepts were generally closer to zero, and $R^2$ are generally closer to unity for the transfer learning compared to traditional learning. The top model yielded a $R^2 = 0.90$ (range: 0.73–0.90 for traditional learning and 0.83–0.90 for transfer learning). The p-values for the transfer learning were consistently smaller than to those of the traditional learning.

The MAEs for the transfer learning are consistently smaller than to those of the traditional learning. The top model yielded a MAE of 8.5% (range: 17.2% to 21.3% for traditional learning,

**AI-predicted *versus* radiologist severity scores**

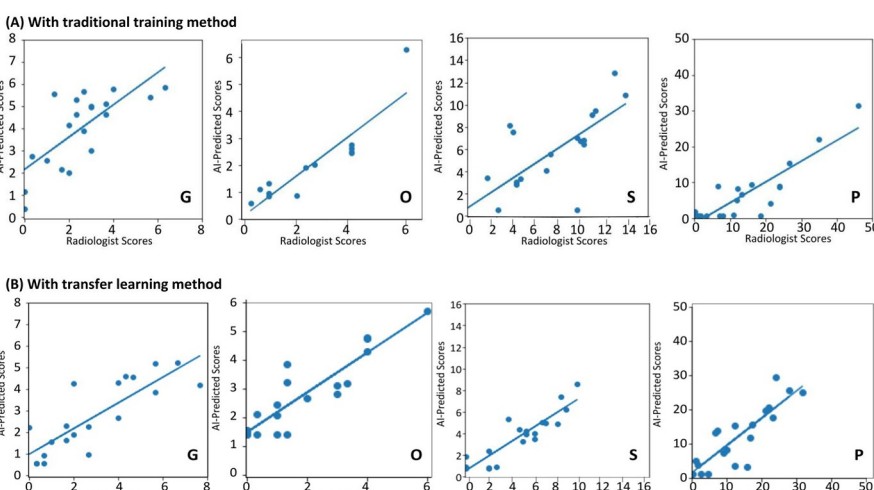

**Fig 5.** Scatterplots of AI-predicted versus radiologist severity scores on CXR using (A) traditional training method by CNN and (B) transfer learning method by CNN. G: geographic extent, O: degree of opacity, S: sum of the two score types, P: product of the two score types. The lines are linear regressions. See Table 2 for slopes, intercepts, $R^2$, and p values.

and 8.5% to 15.5% for transfer learning). Transfer learning generally performed better than those traditional learning.

## Discussion

This study tested the hypothesis that deep-learning convolutional neural networks accurately stage disease severity on portable chest x-rays using radiologists' severity scores as ground truths associated with COVID-19 lung infection. Disease severity scores by three radiologists yielded good inter-rater agreement. AI-predicted scores were highly correlated with radiologist scores. Transfer learning improved efficiency and shorten computational time without compromising performance.

Although there are already over 5500 publications on COVID-19 on PubMed (keyword "COVID-19" Apr 21, 2020), only a few studies have reported AI applications to classify COVID-19 versus non-COVID-19 lung infection on CXR [19–22] and CT [23–26]. All of these studies (including many non-peer-reviewed pre-prints cited here) aimed to classify COVID-19 versus non-COVID-19 images, not to stage disease severity. A major challenge for

**Table 2. Summary of slope, intercepts, $R^2$, and p values of the correlation analysis, and the Mean Absolute Error (MAE) of AI-predicted and radiologist scores with traditional and transfer learning.** The % was obtained by dividing MAE by the corresponding maximum score.

|  |  | Slope | Intercept | $R^2$ | p-value | MAE (%) |
|---|---|---|---|---|---|---|
| Traditional learning | Geographic extent score (0–8) | 0.73 | 2.16 | 0.73 | 0.02 | 1.51 (18.9%) |
|  | Opacity score (0–6) | 0.77 | 0.07 | 0.90 | 0.001 | 1.26 (21.0%) |
|  | Sum (0–14) | 0.67 | 0.71 | 0.76 | 0.0003 | 2.57(18.4%) |
|  | Product (0–48) | 0.58 | -1.60 | 0.88 | 0.06 | 8.26 (17.2%) |
| Transfer learning | Geographic extent score (0–8) | 0.60 | 0.99 | 0.83 | <0.0001 | 0.91(11.4%) |
|  | Opacity score (0–6) | 0.84 | 1.48 | 0.87 | <0.0001 | 0.93 (15.5%) |
|  | Sum (0–14) | 0.68 | 0.45 | 0.90 | 0.001 | 2.12 (15.1%) |
|  | Product (0–48) | 0.80 | 1.37 | 0.85 | <0.0001 | 4.10 (8.5%) |

AI in the application to disease diagnosis is that training datasets were limited to a few lung diseases, resulting in low generalizability. Amongst these studies, most compare CXR of COVID-19 infection with those of bacterial pneumonia and/or normal CXR. More importantly, AI diagnosis of disease in general, and COVID-19 infection in particular, based on CXR has poor specificity because many pathologies have similar radiographic appearance as other infections and diseases on CXR. Definitive diagnosis of COVID-19 infection still requires a test by reverse transcription polymerase chain reaction of a nasopharyngeal or oropharyngeal swab.

Our study differs from these previous studies in that we used AI of CXR to stage disease severity on a graded scale in positive COVID-19 patients. Deep-learning AI, specifically a convolutional neural network, is well suited to extract information from CXR and stage disease severity by training using chest radiologist determination of disease severity scores. We evaluated individual and combination scores of severities as it is unknown which type of scores more accurately reflect lung infection disease severity. Our findings showed that most scores yielded similar correlation coefficients. One possible explanation is that both the opacity score and geographic extent yielded similar information regarding disease severity. Further studies are needed to determine which is superior. It is worth noting that performance by receiver-operating curve (ROC) analysis (such as area under the curve (AUC), accuracy, sensitivity and specificity) cannot be used for continuous variables. Instead, correlation analysis and mean square error analysis were performed.

There were strong correlations between the AI-predicted and radiologist scores, with the top model yielding a $R^2 = 0.90$ and a mean absolute error of 8.5%. This is remarkable performance. The ideal intercept should be zero and ideal slope should be unity. The intercepts were generally close to zero. However, all the slopes were consistently below the line of unity. A possible explanation is that the score distributions were skewed by low score values. As the models attempted to minimize loss, it tended to underestimate higher scores, resulting in slope less than unity. Other explanations are possible. Further studies are needed.

Another novelty of our study is that we compared traditional and transfer learning. Transfer learning approach is expected to yield good performance with small datasets. Most of the performance indices (such as $R^2$, p values, and MAEs) were better with transfer learning compared to traditional learning. The training time with transferring learning was shorter without compromising performance.

Although radiology reports are informative, they are usually qualitative. A quantitative score of disease severity of portable CXR afforded by AI in an accurate and efficient manner should prove useful in clinical settings of COVID-19 circumstance. With CXR being an indispensable imaging tool in the management of COVID-19 lung infection, radiologists need an efficient and accurate means to stage disease severity and monitor disease progression and treatment response. Furthermore, with the anticipated widespread shortage of ICU beds and ventilators, frontline physicians need an efficient and accurate means to triage patients, assess risk and allocate resources. This AI approach to stage disease severity is ideally suited to tackle these challenges associated with the COVID-19 pandemics or the like.

This proof-of-concept pilot study has several limitations. First, this retrospective study was conducted on an open multi-institutional dataset with a small sample size. This topic is timely and only small dataset are currently available at the time of this writing. These findings need to be replicated on larger sample size with multi-institutional data. It may also be of important to investigate multiple longitudinal time points to evaluate disease progression. Second, we did not study the correlation of radiographic score severity with clinical disease severity or clinical outcome which were unavailable. Although radiographic score has been widely used as a surrogate marker of disease severity in a variety of lung diseases (see review [4]), it is unknown at

this time if radiographic scoring reflects functional or clinical outcome in the case of COVID-19. Third, a variety of radiographic scoring systems has been used in other studies [5, 28], each has its advantages and disadvantages. A more sophisticated score system could be explored. Future studies should also consider incorporating non-imaging data (such as demographics, co-morbidities, vitals, blood biomarkers).

In conclusion, deep-learning convolutional neural networks accurately stage lung disease severity on portable chest x-rays associated with COVID-19 lung infection. This approach has the potential to be used to prognosticate, stage disease severity, monitor disease progression and treatment response, which in turn could inform risk management and resource allocation associated with the COVID-19 pandemic.

## Author Contributions

**Conceptualization:** Jocelyn Zhu, Tim Q. Duong.

**Data curation:** Jocelyn Zhu, Beiyi Shen, Almas Abbasi, Mahsa Hoshmand-Kochi, Haifang Li.

**Formal analysis:** Jocelyn Zhu, Beiyi Shen, Almas Abbasi, Mahsa Hoshmand-Kochi, Haifang Li.

**Investigation:** Haifang Li, Tim Q. Duong.

**Methodology:** Jocelyn Zhu.

**Supervision:** Tim Q. Duong.

**Validation:** Jocelyn Zhu.

**Writing – review & editing:** Jocelyn Zhu, Beiyi Shen, Haifang Li, Tim Q. Duong.

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
