## [Decision Letter · Decision Letter 0]

5 Jun 2020

PONE-D-20-16042

Deep learning artificial intelligence accurately stages COVID-19 disease severity on portable chest radiographs

PLOS ONE

Dear Dr. Duong,

Thank you for submitting your manuscript to PLOS ONE. After careful consideration, we feel that it has merit but does not fully meet PLOS ONE’s publication criteria as it currently stands. Therefore, we invite you to submit a revised version of the manuscript that addresses the points raised during the review process.

We look forward to receiving your revised manuscript.

Kind regards,

Dilbag Singh

Academic Editor

PLOS ONE

Additional Editor Comments:

I think authors should add more recent literature from the PLOS One journal.

4. Thank you for stating the following in your Competing Interests section: 'No'

a. Please complete your Competing Interests statement to state any Competing Interests. If you have no competing interests, please state "The authors have declared that no competing interests exist.", as detailed online in our guide for authors at http://journals.plos.org/plosone/s/submit-now

Reviewers' comments:

Reviewer's Responses to Questions

**Comments to the Author**

1. Is the manuscript technically sound, and do the data support the conclusions?

Reviewer #1: Yes

Reviewer #2: Yes

Reviewer #3: Yes

2. Has the statistical analysis been performed appropriately and rigorously? 

Reviewer #1: Yes

Reviewer #2: Yes

Reviewer #3: Yes

3. Have the authors made all data underlying the findings in their manuscript fully available?

Reviewer #1: Yes

Reviewer #2: Yes

Reviewer #3: Yes

4. Is the manuscript presented in an intelligible fashion and written in standard English?

Reviewer #1: Yes

Reviewer #2: No

Reviewer #3: Yes

5. Review Comments to the Author

Reviewer #1: 1. There are some grammatical and typo mistakes in the entire manuscript such as:

“predict continuous values. (Figure 1).”

described by Wong et al (7)

2. Use consistent words such as either COVID19 or COVID-19 in entire manuscript.

3. Please write suitable background of Deep-learning convolutional neural network (CNN) and deep transfer learning?

4. Add more mathematical background of CNN.

5. Add suitable diagrammatic flow in the paper.

6. Authors have use CNN or deep transfer learning? If deep transfer learning is used, then title should be improved.

7. Recently, many studies have been used for diagnosis of COVID-19, so, I suggest authors to add a related work about those studies such as: 10.1148/radiol.2020200432,10.1007/s10096-020-03901-z, 10.1142/S0217984919500222, 10.1049/iet-syb.2019.0116, 10.1148/radiol.2020200642, 10.1148/radiol.2020200463, 10.1148/radiol.2020200370

8. Do improve abstract such as kappa is simply used as k. Also, authors should use present sentences rather than was were.

9. Figure 1 should be improved it seems directly copied from some source.

Reviewer #2: In this paper deep-learning convolutional neural networks has been implemented for novel COVID-19 . Problem has a great significance. The content is written well and the results are promising. The paper needs some improvement in terms of the following recommendations:

1. Main objective need to be explained in a specific way.

2. Introduction is week, need to include some more similar modeling approaches and related work on other nonlinear modeling approaches should be extended with the following papers:

Apostolopoulos, I. D., & Mpesiana, T. A. (2020). Covid-19: automatic detection from x-ray images utilizing transfer learning with convolutional neural networks. Physical and Engineering Sciences in Medicine, 1.

Singh, D., Kumar, V., & Kaur, M. (2020). Classification of COVID-19 patients from chest CT images using multi-objective differential evolution–based convolutional neural networks. European Journal of Clinical Microbiology & Infectious Diseases, 1-11.

Rahimzadeh, M., & Attar, A. (2020). A modified deep convolutional neural network for detecting COVID-19 and pneumonia from chest X-ray images based on the concatenation of Xception and ResNet50V2. Informatics in Medicine Unlocked, 100360.

Hall, L. O., Paul, R., Goldgof, D. B., & Goldgof, G. M. (2020). Finding covid-19 from chest x-rays using deep learning on a small dataset. arXiv preprint arXiv:2004.02060.

3. Methodology need to be written in more detail with respect of each step involved. Statistical methods and performance evaluation is also required more information.

4. Results need to be compared with exiting works.

Reviewer #3: This paper is nicely written and idea is novel, but I have following recommendations

The choice of the number of layers and neurons per layer given in section III A is not argued.

The database is weak to draw a conclusion.

Taking into account the uncertainties given in the various analyzes, the results are comparable. In addition, the efficiencies must be revised downwards.

Please explain how this manuscript advances this field of research and/or contributes something new to the literature.

6. PLOS authors have the option to publish the peer review history of their article (what does this mean?). If published, this will include your full peer review and any attached files.

Reviewer #1: No

Reviewer #2: No

Reviewer #3: No

---

## [Author Response · Author response to Decision Letter 0]

10 Jun 2020

Pls see attachment for easier read. 

Dear Reviewers, 

We thank you for your constructive comments. Please find the attached point-by-point response below. We believe we have addressed all comments positively. We have also carefully edited throughout. We have revised/removed statement of “data not shown.” Note that we also added the mean absolute error as another measure. 

Reviewer #1: 

1. There are some grammatical and typo mistakes in the entire manuscript such as:

“predict continuous values. (Figure 1).”

described by Wong et al (7)

Thank you. Corrected. It was incorrectly stated, it should have been Wang et al. 

2. Use consistent words such as either COVID19 or COVID-19 in entire manuscript.

Thank you. Corrected.

3. Please write suitable background of Deep-learning convolutional neural network (CNN) and deep transfer learning?

The following is added. 

Deep-learning artificial intelligence (AI) has become increasingly popular for analyzing diagnostic images [10, 11]. AI has the potential to facilitate disease diagnosis of disease, diseases severity staging, and longitudinal monitoring of disease progression. One common machine-learning algorithm is the convolutional neural network (CNN) [12, 13], which takes an input image, learns important features in the image such as size or intensity, and saves these parameters as weights and bias to differentiate types of images [14, 15]. CNN architecture is ideally suited for analyzing images. 

The majority of machine learning algorithms to date are trained to solve specific tasks, working in isolation. Models have to be rebuilt from scratch if the feature-space distribution changes. Transfer learning overcomes the isolated learning paradigm by utilizing knowledge acquired for one task to solve related ones. Transfer learning in AI is particularly important for small sample size data because the pre-trained weights enables more efficient training and better performance [16, 17]. 

There are many AI algorithms that have already been developed for CXR applications (see review [18]) and these AI algorithms can be repurposed to study COVID-19 lung infection. A few studies have already reported AI applications to classify COVID-19 versus non-COVID-19 lung infection using CXR [19-22] and CT [23-26]. Many of these publications are pre-print, non-peer reviewed publications. These studies aimed to classify (not stage) COVID-19 versus non-COVID-19 images. Classification (i.e., for the purpose of diagnosis) of COVID-19 versus non-COVID-19 made by AI unfortunately has poor specificity. The reverse-transcription polymerase chain reaction test of a nasopharyngeal or oropharyngeal swab is still needed for definitive diagnosis. By contrast, deep-learning AI is well suited to stage disease severity. To our knowledge, there have been no studies to date that use deep-learning AI to stage disease severity on CXR of COVID-19 lung infection. 

The goal of this study was to employ deep-learning CNN to stage disease severity of COVID-19 infection on CXR (as opposed to classify COVID-19 versus non-COVID-19 lung infection). The ground truths were the disease severity scores determined by expert chest radiologists. Comparison was made with traditional and transfer learning. This approach has the potential to provide frontline physicians an efficient and accurate means to triage patients, assess risk, allocate resources, as well as to monitor disease progression and treatment response. This approach is timely and urgent because of the anticipated widespread shortage of ICU beds and mechanical ventilators. 

4. Add more mathematical background of CNN.

This paper deals with an innovative CNN application to stage lung disease severity based on portable CXRs of COVID-19 patients. The CNN algorithm and transferring methods are standard. We feel that to add a mathematical background of CNN would be repeating as the literature has already described it in a much more eloquent and precise manner. We only provide a brief qualitative description and cited references instead. 

5. Add suitable diagrammatic flow in the paper.

A flowchart diagram for analysis is added.

6. Authors have use CNN or deep transfer learning? If deep transfer learning is used, then title should be improved.

The new title is: 

Deep transfer learning methods in artificial intelligence accurately stages COVID-19 lung disease severity on portable chest radiographs

7. Recently, many studies have been used for diagnosis of COVID-19, so, I suggest authors to add a related work about those studies such as: 10.1148/radiol.2020200432, 10.1007/s10096-020-03901-z, 10.1142/S0217984919500222, 10.1049/iet-syb.2019.0116, 10.1148/radiol.2020200642, 10.1148/radiol.2020200463, 10.1148/radiol.2020200370

Thank you for the references. They are cited. Our study employed deep learning to stage disease severity of COVID-19 infection on CXR (as opposed to classify COVID-19 versus non-COVID-19 lung infection that are in the literature). To our knowledge, there have been no studies to date that use deep-learning AI to stage disease severity on CXR of COVID-19 lung infection.

8. Do improve abstract such as kappa is simply used as k. 

Also, authors should use present sentences rather than was were.

Thank you. ��is used more often in the clinical literature so we spelled it out instead. We changed to present tense where appropriate but past tense is used more often in the clinical literature. 

9. Figure 1 should be improved it seems directly copied from some source.

This figure was created by us. It was not copied.

Reviewer #2: In this paper deep-learning convolutional neural networks has been implemented for novel COVID-19. Problem has a great significance. The content is written well and the results are promising. The paper needs some improvement in terms of the following recommendations:

1. Main objective need to be explained in a specific way.

Thank you. Pls see the response to Reviewer 1, comment #3 for the extensively revised Introduction. 

2. Introduction is week, need to include some more similar modeling approaches and related work on other nonlinear modeling approaches should be extended with the following papers:

Apostolopoulos, I. D., & Mpesiana, T. A. (2020). Covid-19: automatic detection from x-ray images utilizing transfer learning with convolutional neural networks. Physical and Engineering Sciences in Medicine, 1.

Singh, D., Kumar, V., & Kaur, M. (2020). Classification of COVID-19 patients from chest CT images using multi-objective differential evolution–based convolutional neural networks. European Journal of Clinical Microbiology & Infectious Diseases, 1-11.

Rahimzadeh, M., & Attar, A. (2020). A modified deep convolutional neural network for detecting COVID-19 and pneumonia from chest X-ray images based on the concatenation of Xception and ResNet50V2. Informatics in Medicine Unlocked, 100360.

Hall, L. O., Paul, R., Goldgof, D. B., & Goldgof, G. M. (2020). Finding covid-19 from chest x-rays using deep learning on a small dataset. arXiv preprint arXiv:2004.02060.

Pls see the response to Reviewer 1, comment #3 for the extensively revised Introduction. 

Thank you for the references. They are cited. 

3. Methodology need to be written in more detail with respect of each step involved. Statistical methods and performance evaluation is also required more information.

Thank you. We have expanded the Methods in multiple places and included a flowchart. 

4. Results need to be compared with exiting works.

Thank you. We expanded the discussion of the results generally. There are currently no studies to date that use deep-learning AI to stage disease severity on CXR of COVID-19 lung infection. There are no similar works to compare quantitatively. 

Reviewer #3: This paper is nicely written and idea is novel, but I have following recommendations

The choice of the number of layers and neurons per layer given in section III A is not argued.

Thank you. The following is added.

CNN: A convolutional neural network (Figure 2A)[12, 13] with an additional regression layer was utilized to predict disease severity scores of CXR on a graded scale. The images were normalized, converted to RGB images, resized to 64x64 pixel images, and separated into 80% training and 20% testing datasets. This model architecture included Convolutional, Activation, Batch Normalization, Max Pooling, Dense layers, and a final Dense regression layer to fit the model to predict continuous values. Specifically, for convolutional neural networks, deep learning models are well-suited to learn complex functions, and optimize performance of regularization methods. Batch normalization layers served to stabilize the learning process by standardizing inputs, and standard rectified linear activation layers for CNNs were used to optimize performance. In general, the number of layers and nodes depends heavily on the input data and prediction goals, therefore heuristics were used to generate a standard CNN model for image analysis, the model was then fine-tuned based on performance on the testing dataset. This resulted in three 3x3 kernel sized convolutional layers. L2 regularization and Dropout layers, using a parametric probability of 0.3 as determined by analyzing performance results, were used to prevent overfitting. An optimal batch size of 8 was determined by five-fold cross validation, and the model was trained for 110 epochs. The loss function was measured by mean squared error as the model predicted continuous values, and the learning rate was set to 0.001, as it proved to perform more positively than the standard 0.01 rate in early training, and the Adam optimizer proved the most successful in minimizing loss.

The database is weak to draw a conclusion.

We agree that the available CXR data are of small sample size and have limited information. We emphasized these in the Limitations section. We hope that future studies will include larger multi-institutional dataset. 

Taking into account the uncertainties given in the various analyzes, the results are comparable. In addition, the efficiencies must be revised downwards.

Thank you. We toned down efficiency claim. 

Please explain how this manuscript advances this field of research and/or contributes something new to the literature.

Thank you. We have revised the last paragraph of the Introduction and Discussion to reflect your comments.

---

## [Decision Letter · Decision Letter 1]

13 Jul 2020

Deep transfer learning artificial intelligence accurately stages COVID-19 lung disease severity on portable chest radiographs

PONE-D-20-16042R1

Dear Dr. Duong,

We’re pleased to inform you that your manuscript has been judged scientifically suitable for publication and will be formally accepted for publication once it meets all outstanding technical requirements.

Kind regards,

Dilbag Singh

Academic Editor

PLOS ONE

Additional Editor Comments (optional):

Reviewers' comments:

Reviewer's Responses to Questions

**Comments to the Author**

1. If the authors have adequately addressed your comments raised in a previous round of review and you feel that this manuscript is now acceptable for publication, you may indicate that here to bypass the “Comments to the Author” section, enter your conflict of interest statement in the “Confidential to Editor” section, and submit your "Accept" recommendation.

Reviewer #1: All comments have been addressed

Reviewer #2: All comments have been addressed

2. Is the manuscript technically sound, and do the data support the conclusions?

Reviewer #1: Yes

Reviewer #2: Yes

3. Has the statistical analysis been performed appropriately and rigorously? 

Reviewer #1: Yes

Reviewer #2: Yes

4. Have the authors made all data underlying the findings in their manuscript fully available?

Reviewer #1: Yes

Reviewer #2: Yes

5. Is the manuscript presented in an intelligible fashion and written in standard English?

Reviewer #1: Yes

Reviewer #2: Yes

6. Review Comments to the Author

Reviewer #1: (No Response)

Reviewer #2: The recommendations given previously have been addressed successfully.

The paper is Accepted as it is.

7. PLOS authors have the option to publish the peer review history of their article (what does this mean?). If published, this will include your full peer review and any attached files.

Reviewer #1: No

Reviewer #2: No

---

## [Editor Report · Acceptance letter]

20 Jul 2020

PONE-D-20-16042R1 

Deep transfer learning artificial intelligence accurately stages COVID-19 lung disease severity on portable chest radiographs 

Dear Dr. Duong:

I'm pleased to inform you that your manuscript has been deemed suitable for publication in PLOS ONE. Congratulations! Your manuscript is now with our production department. 

Kind regards, 

on behalf of

Dr. Dilbag Singh 

Academic Editor

PLOS ONE